# Structural Factors That Determine the Activity of the Xenobiotic Reductase B Enzyme from *Pseudomonas putida* on Nitroaromatic Compounds

**DOI:** 10.3390/ijms24010400

**Published:** 2022-12-26

**Authors:** Manuel I. Osorio, Nicolás Bruna, Víctor García, Lisdelys González-Rodríguez, Matías S. Leal, Francisco Salgado, Matías Vargas-Reyes, Fernando González-Nilo, José M. Pérez-Donoso, Osvaldo Yáñez

**Affiliations:** 1Center for Bioinformatics and Integrative Biology, Facultad de Ciencias de la Vida, Universidad Andres Bello, Av. República 330, Santiago 8370146, Chile; 2Centro de Investigación Biomédica, Facultad de Medicina, Universidad Diego Portales, Ejército 141, Santiago 837007, Chile; 3Facultad de Química e Ingeniería Química, Universidad Nacional Mayor de San Marcos, Lima 15081, Peru; 4Facultad de Ingeniería y Negocios, Universidad de las Américas, Sede Concepción, Santiago 9170022, Chile; 5Núcleo de Investigación en Data Science, Facultad de Ingeniería y Negocios, Universidad de las Américas, Santiago 7500000, Chile

**Keywords:** molecular dynamics simulation, quantum mechanics, substrate promiscuity of enzymes, nitroaromatic compound reactivity, protein–ligand interactions

## Abstract

Xenobiotic reductase B (XenB) catalyzes the reduction of the aromatic ring or nitro groups of nitroaromatic compounds with methyl, amino or hydroxyl radicals. This reaction is of biotechnological interest for bioremediation, the reuse of industrial waste or the activation of prodrugs. However, the structural factors that explain the binding of XenB to different substrates are unknown. Molecular dynamics simulations and quantum mechanical calculations were performed to identify the residues involved in the formation and stabilization of the enzyme/substrate complex and to explain the use of different substrates by this enzyme. Our results show that Tyr65 and Tyr335 residues stabilize the ligands through hydrophobic interactions mediated by the aromatic rings of these aminoacids. The higher XenB activity determined with the substrates 1,3,5-trinitrobenzene and 2,4,6-trinitrotoluene is consistent with the lower energy of the highest occupied molecular orbital (LUMO) orbitals and a lower energy of the homo orbital (LUMO), which favors electrophile and nucleophilic activity, respectively. The electrostatic potential maps of these compounds suggest that the bonding requires a large hydrophobic region in the aromatic ring, which is promoted by substituents in ortho and para positions. These results are consistent with experimental data and could be used to propose point mutations that allow this enzyme to process new molecules of biotechnological interest.

## 1. Introduction

The enzyme xenobiotic reductase B (XenB) catalyzes the transfer of a hydride ion from the cofactor FMN to various substrates of clinical, industrial or biotechnological interest [1]. The study of this enzyme will allow a better understanding of its ability to recognize and process different substrates, as well as provide background information that will allow projecting this knowledge into new biotechnological applications [1,2,3,4,5]. However, the poor functional and structural understanding of this catalytic ability has limited the potential new applications of XenB.

The ability of an enzyme to process different substrates has been termed promiscuity and is a property identified in several enzymes [6]. It is a relevant ability since it could be associated with the evolution of proteins. For the case of chemically related substrates, such as the case we will address in this article, it has been called substrate ambiguity. Thus, when we speak in a general way about enzymes that can process different substrates, we will say substrate promiscuity and when we refer to the special case we are dealing with in this article we will say substrate ambiguity.

In general—and probably valid for XenB—it has been proposed that the substrate promiscuity of enzymes is determined by the structural properties of the active site of the protein, analogous to substrate specificity. The structural mechanisms underlying this enzymatic capacity can be explained on the basis of the conformational diversity of the enzyme active site, different protonation states, different subsites within the active site and cofactor ambiguity [6]. Enzymes, such as B-lactamases, sulfotransferases or isopropylmalate isomerase, present more than one conformation capable of binding substrates. This adaptation can be achieved by several small conformational changes or by the movement of a single region of the protein [7,8,9]. Similarly, a region of the active site of XenB could be flexible enough to accommodate different ligands through conformational changes that allow binding. Another example is cytochrome P450 or cytochrome 3A4, which can accommodate different substrates as they have several residues that could be involved in ligand binding to the substrate binding site [10,11]. Since XenB catalyzes substrate reduction by transferring a hydride ion from a single cofactor Flavin mononucleotide (FMN), the substrate ambiguity and even substrate promiscuity of this enzyme could depend on the conformational flexibility of the substrate binding site, different protonation states or different subsites within the active site.

XenB can reduce the nitro groups or the aromatic ring of 2,4,6-trinitrotoluene (TNT) or 2,4,6-trinitrophenol (TNF). It can act on aromatic and aliphatic substrates, with different functional groups: TNT, TNF, nitroglycerin, trinitroxylene (TNX), hexahydro−1-nitroso-3,5-dinitro-1,3,5-triazine (MNX), 2,6-dinitrotoluene, 2,4-dinitrobenzene, 1,3, 5-trinitrobenzene, 4-amino-2, 6-dinitrotoluene, cyclohexen-2-one (CHE), N-ethylmaleimide (NEM), 5-[bis-2(chloroethyl)-amino]-2,4-dintrobenzamide (SN23862) or 5-(aziridin-1-yl)-2,4-dinitrobenzamide (CB1954) [3,12,13,14]. These compounds are characterized by having nitro groups or conjugated double bonds, which are necessary for XenB to process the substrate but are not sufficient to ensure catalysis. Except for CHE and NEM, all identified substrates of the enzyme contain reducible nitro groups. However, the presence of this group does not ensure the activity of the enzyme on this compound (Table 1). Despite the substrate promiscuity of XenB, this enzyme cannot act on chemically related compounds. XenB is inactive in presence of 2,6-dinitrotoluene (DN4) despite its chemical similarity to TNT or 2,4-dinitrotoluene (DN6). It also reduces 4-amino-2,6-dinitrotoluene (4AD), but is inactive for 2-amino-4,6-dinitrotoluene (2AD). It can also reduce 2,4-dinitrobenzene and 1,3,5-trinitrobenzene (TNB) but not dinitrobenzene (1,2-dinitrobenzene, 1,3-dinitrobenzene, or 1,4-dinitrobenzene) or nitrobenzene (NBZ).

The structural factors that determine the substrate promiscuity of XenB, allowing it to process compounds of different structure or to discriminate between chemically similar nitro compounds, are unknown. Some structural information on the interaction of XenB with TNT, TNF, nitroglycerin or biomimetic analogues of NADPH has been obtained through molecular dynamics simulations [15,16]. In addition, there is experimental data on XenB activity with several aromatic nitrocompounds, including the prodrug (CB1954) [5]. In the case of TNT, XenB establishes a dynamic interaction involving regions of the protein close to the substrate binding site (odd loops) that favors FMN/TNT coplanarity, a conformation necessary for the transfer of the hydride ion to the aromatic ring [17]. TNF maintains interactions with the same region as TNT but adopts two conformations, one that brings the nitro group attached to carbon 4 closer and another that moves this group away from the cofactor [17]. A theoretical study on the reduction of TNT and the Meisenheimer complex generated by the transfer of a hydride to the aromatic ring of TNT provided information on the selectivity for the aromatic ring [18]. This study suggests that an increase in the negative charge of the substrate could favor the reduction of the aromatic ring over the nitro groups, a process in which the conserved residues His172 or Asn175 could participate [18,19].

Considering the environmental impact of XenB-processed substrates and their potential application in prodrug design, in the present work, we evaluated the structural factors determining the interaction of XenB with TNT, DN4, DN6, 4AD, 2AD, TNB and NBZ.

## 2. Results and Discussion

In previous works, it has been experimentally demonstrated that XenB presents enzymatic activity with the benzene compounds TNT, DN6, 4AD and TNB but not in the presence of DN4, 2AD or NBZ [13]. All of these compounds contain a six-carbon aromatic ring and at least one nitro group; however, the enzyme shows relevant differences in activity compared to all of them. XenB shows four times more activity in presence of TNB than TNT. In contrast, XenB activity with TNT is 4 and 25 times higher than that observed with 4AD and DN6, respectively (Table 1). This ability to discriminate between similar compounds could be related to the flexibility of the substrate binding site, different conformations adopted by the enzyme when binding to each ligand or to the chemical reactivity of each substrate.

To evaluate the participation of specific aminoacidic residues in the formation of the enzyme/ligand complex, the non-covalent interactions between different ligands and XenB were characterized by quantum mechanics (QM) calculations. The general reactivity of each compound and its charge distribution (restrained electrostatic potential or RESP) can be used to explain the experimental activities determined for XenB in the presence of the different substrates. However, it is possible that a difference in the stability of the aromatic ring contributes to the lower XenB activity determined with some compounds. To evaluate this factor, the current intensity of the ring was calculated by QM, since a higher current is associated with a higher aromaticity. According to these quantum calculations, we can identify differences in charge distribution, the feasibility of receiving electrons, or ring stability and, based on these characteristics, explain the differences in XenB activity observed with these compounds.

### 2.1. Molecular Dynamics Simulations

According to reports by other researchers, XenB can act on TNT, DN6, 4AD and TNB but not on DN4, 2AD or NBZ (Table 1). To identify whether this activity is associated with the formation of a stable XenB/ligand complex, molecular dynamics simulations were performed. As observed in the simulations (Figure 1), XenB forms stable complexes with TNT, DN6, 4AD and TNB, all substrates on which the enzyme shows enzymatic activity. The simulations also show that XenB forms a stable complex with 2AD even though it shows no activity on this ligand (Figure 2), suggesting that it could be a competitive inhibitor of XenB. In contrast, it does not form stable complexes with DN4 and NBZ, which is consistent with the inactivity of the enzyme on them (Figure 2).

It has been shown that the activation energy of the hydride transfer from the FMN cofactor to TNT increases with the distance between the hydride acceptor and the donor group (HAD distance). In accordance with this observation, higher enzyme activity should be correlated with smaller HAD distances. As shown in Figure 1, the functional groups closest to the cofactor are slightly closer to TNT than to DN6, which is consistent with the lower XenB activity in this compound. In contrast, the HAD distance for TNB is greater than for TNT, which is not consistent with the 4-fold greater activity for TNB. This higher activity for TNB could be explained based on the chemical reactivity of this compound, which will be discussed in Section 2.2 below.

#### Trajectory and Cluster Analysis

Molecular dynamics simulations of 200 ns were performed, recording 50 frames every 1 ns (10,000 frames in total). According to the simulations, the XenB/ligand complex adopts a stable conformation that can be identified by cluster analysis (Figure 1). Using this statistical methodology, it is possible to analyze the trajectories and identify similar conformations to group them into clusters. To perform the clustering, the density-based spatial method of applications with noise (DBSCAN) implemented in the CPPTRAJ tool was used [20]. This algorithm considers a cluster as a contiguous region of high point density, separated from other similar clusters by contiguous regions of low point density. Each analysis was performed with a cutoff distance of 1.5 Å based on the root mean square deviation (RMSD), without considering the ligand hydrogen atoms, and for at least five points for each cluster. According to this calculation, representative structures were obtained for each simulation (Figure 3) in relation to the position of the ligand in the protein.

The enzymatic reaction catalyzed by XenB requires the binding of the substrate to the substrate binding site and the generation of a stable complex. Figure 3 shows the representative conformations from the 200 ns molecular dynamics simulation for the enzyme/ligand complexes that can be processed by XenB: TNT (Figure 3A), DN6 (Figure 3B), 4AD (Figure 3C), DN4 (Figure 3D) and TNB (Figure 3E,F). These representative conformations were analyzed by QM calculations to identify the nature of the non-covalent interaction index (NCI), which allows describing this type of interaction between the protein and each ligand (bottom panels of Figure 3). As observed in this analysis, the Tyr 65 residue remains within 3 Å of the substrates TNT (A), DN6 (B), 4AD (C), DN4 (D) and TNB (E, F). The Leu129 residue remains at a short distance (less than 3 Å) from TNT, DN4 and TNB, establishing weak interactions with these substrates (green marked in the lower panels). These interactions could be relevant in an aqueous medium, since they could stabilize the hydrophobic regions of the substrate, removing water molecules from the hydration layer of the compounds. In the case of Tyr65, the aromatic ring allows the establishment of a hydrophobic surface large enough to maintain the enzyme/ligand binding with TNT (A) and DN6 (B). Moreover, these hydrophobic interactions maintain a coplanar conformation of the aromatic ring of the ligands and the FMN cofactor, which promotes a shorter HAD distance favoring hydride transfer. Residues Tyr 177 and Thr24 could interact with nitro groups; however, the nature of the NCIs does not reveal dipole/dipole interactions or hydrogen bridges. Although DN6 and TNT adopt similar conformations (Figure 3A,B), the activity of XenB by DN6 is lower (4% of that of TNT, Table 1). These differences in enzyme activity, for substrates with similar conformations, could be explained by the reactivity of the different substrates, which will be addressed in the next section. For the 4AD (C) ligand, Tyr65 could stabilize the substrate by favoring a conformation that brings a nitro group closer to the cofactor, an interaction in which Trp99 could also participate. This conformation of 4AD is consistent with the activity shown by the enzyme on this compound (Figure 1E and Figure 3C), reaching 25% of the activity on TNT (Table 1). The conformations adopted by TNB (Figure 3E,F) and DN4 (Figure 3D) are stabilized by Van der Waals interactions with Tyr65 and Tyr335, keeping one of the nitro groups close to the cofactor. However, the conformation adopted by TNB and DN4 does not explain the lower XenB activity by DN4 (0.4% of that of TNT) nor the high activity of TNB (four times that of TNT).

### 2.2. Quantum Chemical Reactivity Indexes

As observed in quantum mechanical calculations, the ligands are stabilized at the substrate binding site by non-covalent interactions. These Van der Waals bonds hold the ligands at distances that allow the transfer of the hydride ion from the cofactor to the substrate. However, this distance (Figure 1) or conformation (Figure 3) does not allow explaining the relative difference between the activity of XenB by TNT with respect to the compounds DN6 (4), DN4 (0.4) and TNB (400). Interestingly, despite the coplanar orientation of the aromatic ring of TNT and DN6 with respect to the FMN cofactor, XenB exhibits lower activity by DN6. In this sense, the conformation adopted by DN4 and TNB, or the enzyme/ligand interactions established, cannot explain the higher activity of XenB with TNB. Depending on the presence of aromatic ring deactivating (methyl) or activating (amino) groups, the substrates may have different reactivity. To account for these differences in activity, QM calculations were performed to evaluate substrate reactivity based on global reactivity indices, HOMO/LUMO orbitals, charge density and aromatic ring currents.

#### 2.2.1. Global Reactivity Molecular Descriptors

For the model studied, it is observed that ligand binding is necessary but not sufficient for XenB activity, as observed for XenB/2AD, which remains bound in the simulation but does not show activity. Another factor that could determine the activity of XenB on a substrate is the reactivity of each ligand. Thus, for example, the higher activity of XenB on TNB than for TNT could be associated with a different reactivity.

According to this analysis, to predict the reactivity of ligand molecules against XenB, chemical reactivity descriptors based on density functional theory were evaluated by calculating global reactivity indices, approximated by Koopmans theorem. The following indices were obtained: electronegativity (χ), global hardness (η) electrophilicity (ω), electrodonation (ω^−^), electroacceptance (ω^+^) and net electrophilicity (Δω^±^) for the seven nitroaromatic compounds (TNT, DN6, DN4, 4AD, 2AD, TNB and NBZ) for which we have experimental data of enzymatic activity (Table 2) [21]. In general terms, TNB, TNT and DN6 compounds have high electronegativity values, with TNB and TNT having the highest electronegativity values among the seven nitramine compounds. However, systems with high electronegativity values could favor the electron delocalization of the hydride ion, favoring the stabilization of the first stage of the reaction. On the other hand, 4AD and 2AD have lower hardness values than the other molecules, so they become less hard, less opposed to electron flow and, therefore, are more reactive systems. In the case of electrophilicity values, TNT and TNB have the highest values compared to the values of the other molecules, indicating that these molecules are more electrophilic and therefore more likely to react with the hydride ion that could be transferred from the cofactor.

The *p*-boundary orbitals provide information about the energies of the highest occupied molecular orbital (HOMO) and the lowest unoccupied molecular orbital (LUMO), which can act as electron donors and acceptors, respectively [22]. The boundary orbital gives an idea of the reactivity of the molecule, and the active site can be demonstrated by the distribution of the boundary orbital. Thus, the higher the HOMO energy, the higher the ability to perform nucleophilic attacks, i.e., donate electrons; it is an important electronic parameter to describe the mode of interaction of drugs with other molecules, such as the interactions between these drugs and their receptors. Table 2 and Figure 4 show the values of HOMO, LUMO energy and their difference. Note that the order of the energy gap is TNB > NBZ > DN6 == DN4 > TNT > 4AD == 2AD. As the energy gap decreases, the reactivity of the molecule increases, and it can be easily excited with low energy. Surprisingly, the molecules 4AD and 2AD have the methyl group and the primary amine group; having lower energy gap values, 4AD and 2AD would be less favorable to a reduction process. The energy level of HOMOs is different for all investigated nitroaromatic compounds. TNB, TNT and DN6 showed a more negative HOMO than the other compounds and, consequently, could be better electron-donor molecules. Interestingly, TNB of the highest energy gap $\Delta \epsilon_{HOMO-LUMO}$ = 5.9 eV, and there are several hydrophilic interactions that could facilitate receptor binding. This suggests that such hydrophilic interactions greatly influence the binding affinity of such small molecules to receptors. The HOMO of a given molecule and the LUMO with adjacent residues could share orbital interactions during the binding process.

#### 2.2.2. Molecular Electrostatic Potential Maps

To show the electrostatic representation of the molecules, electrostatic potential maps (EPM) were developed (Figure 5). These are useful tools for predicting physicochemical properties and provide information on the form and nature of the electrostatic force when it is positive, negative and neutral. EPMs provide a proximate description for assessing the reactivity of these compounds to electrophilic and nucleophilic attack during interaction with other molecules. The positive regions (blue/green) indicate the portion of the molecule with the higher probability of suffering a nucleophilic attack, while the negative regions (green/red) indicate the regions that will perform nucleophilic attacks [23], that is, the chemical reactivity of molecules, since regions of negative potential are expected to be sites of protonation and nucleophilic attack, while regions of positive potential may indicate electrophilic sites. The EPM for TNT, DN6, DN4, 4AD, 2AD, TNB and NBZ show negative zones (red color) of electrostatic potential, in the position of the nitro group (NO_2_), which are favored by electrophilic attacks. In the case of 4AD, 2AD, DN4, DN6 and NBZ, the aromatic rings present a slight green zone, which can facilitate aromatic weak interactions. It is worth mentioning that, in the case of TNB and TNT, the oxygen atom of the nitro group attracts electrons and generates electronic deficiency in the aromatic ring, favoring nucleophilic attack.

#### 2.2.3. Ring Current Densities Calculations

Using the ring current strength (RCS) and vector diagram visualization (Figure 6), we can measure the six-electron π delocalization or aromaticity character for the studied nitroaromatic compounds. Indeed, the diatropic contribution (in blue) is used to assign a positive sign and the paratropic part (in red), the negative sign. Benzene has been chosen as a system of reference, in which the integrations computed from the center of the ring until approximately 5.0 Å outside it and over the rectangular plane bisecting a C-C bond were 17.60 nAT^−1^ and −4.95 nAT^−1^, respectively [24]. Moreover, the difference integration values over rectangular planes from the center of the ring through the atoms is due to different contributions of localized atomic charges [24,25,26,27]. Hence, we conclude net ring current strength calculation for benzene at the PBE0-D3/Def2-TZVP level is 12.7 nAT^−1^ and verified its aromaticity because of the positive sign. The RCS value obtained with the same level of theory shows the aromatic character of nitrobenzene, which is weaker than for benzene, a fact that is deduced from its more positive paratropic ring currents. Therefore, we can compare them and predict a greater aromatic character of 2AD and NBZ with net RCS values close to 12.0 nAT^−1^ with respect to the TNB, TNT and DN6 compound whose net RCS obtained were close to 9.0 nAT^−1^. This similar aromaticity of TNB, TNT and DN6 shows that the aromatic ring of these compounds presents similar reactivity. Compound 4AD has the lowest net RCS value (close to 8.0 nAT^−1^), which makes the ring of this compound more reactive, explaining the catalytic capacity shown by XenB for this compound in relation to 2AD.

## 3. Materials and Methods

### 3.1. Preparation of the Systems

The crystal structure of the P. putida XenB enzyme in complex with TNT and the cofactor FMN was obtained from the Protein Data Bank (PDB) with PDB ID: 4AEO at a resolution of 1.8 Å. The XenB/TNT enzyme model was generated without considering water molecules. For the models of XenB in complex with the other compounds, the Avogadro program was used to modify the coordinates by removing or adding functional groups [28]. For example, to obtain the XenB/DN6 complex, the N and O atoms corresponding to the NO_2_ group attached to carbon atom 6 were eliminated. For 4AD, on the other hand, the N and O atoms of the NO_2_ group attached to carbon 4 were removed, and an NH_2_ group was added at that position. This methodology maintains the distances between the cofactor and substrate functional groups at a similar magnitude to that observed for the enzyme complex with TNT.

### 3.2. Molecular Dynamics (MD) Simulation Protocol

The polypeptide chain of XenB was modeled considering the protonation states of ionizable residues at pH 7.0, determined with the H++ web server [29]. The protein structure was modeled with the ff14SB force field, while the substrate structures were optimized by quantum mechanics with a B3LYP/6–31G* level of theory [30]. The atomic charges (restricted electrostatic potential) of each ligand were calculated with 10 layers of points for each atom with a point density of 2500 and a HF/6–31G* level of theory in the Gaussian 16 program. Finally, a standard MD protocol was followed as follows for the complexes of all models: (1) minimization and structural relaxation of water molecules with 1000 steps of steepest descent minimization, followed by 1000 of steps conjugate gradient minimization, and MD simulation with an NPT ensemble (300 K) for 1000 ps, using harmonic constraints of 10 kcal/(mol∙Å^2^) for protein and ligand; (2) full structure minimization considering 1500 steps of steepest minimization and 500 steps of conjugate gradient minimization; (3) the minimized systems were progressively heated to 300 K with harmonic constraints of 10 kcal/(mol∙Å^2^) on the carbon skeleton and ligand for 0.5 ns; (4) the system was then equilibrated for 0.5 ns maintaining the constraints and then for 5 ns without constraints at 300 K in a canonical ensemble (NVT); and (5) for production, molecular dynamics simulations were run for 200 ns with a 2 fs time step, unconstrained, in an isothermal–isobaric (NPT) ensemble at 300 K. In the MD simulation, temperature was controlled with a Langevin thermostat with a collision frequency of 1 ps^−1^ (NVT) and pressure with a Berendsen barostat (NPT), with 2 ps of relaxion time [17,31,32]. In addition, the particle mesh Ewald (PME) method with a cutoff value of 10 Å was used to treat long-range and bond-free electrostatic interactions [33]. All MD simulation calculations were performed using the Graphics Processing Unit (GPU)-AMBER Implementations 20 [34].

### 3.3. Cluster Analysis

Statistical methodology groups similar data and separates them from data sets with different properties. To perform the clustering, the density-based spatial method of applications with noise (DBSCAN), which is implemented in the CPPTRAJ tool, was used [20,35]. This algorithm performs separation by considering a cluster in the data space as a contiguous region with high point density, separated from other similar clusters by contiguous regions of low point density. Each analysis was performed with a cutoff distance of 1.5 Å based on the root mean square deviation (RMSD) of the different atoms (excluding hydrogen) of the ligand and at least 5 points for each cluster [30]. According to this calculation, representative structures were obtained for each simulation in relation to the position of the ligand in the protein.

### 3.4. Density Functional Theory (DFT) Calculations

Seven nitroaromatic compounds by xenobiotic reductase XenB: 2,4,6 trinitrotolueno (TNT), 2,4, dinitrotolueno (DN6), 2,6 dinitrotolueno (DN4), 4-amino-2,6-dinitrotolueno (4AD), 2-amino-4,6-dinitrotolueno (2AD), 1,3,5-trinitrobenzeno (TNB) and nitrobenzeno (NBZ), were drawn using Discovery Studio 3.1 (Accelrys, CA), and geometry optimizations and vibrational frequency calculations were carried out using the Gaussian16 program [36], using the PBE0 hybrid functional [37] with Grimme’s dispersion correction method D3 [38] in conjunction with the Def2-TZVP [39] basis set. The water was simulated as a solvent using the SMD parametrization of the IEF-PCM. Some DFT-based global reactivity descriptors (Table 3), such as the HOMO–LUMO gap, electronegativity (χ), global hardness (η), electrophilicity (ω), electrodonating (ω^−^) and electroaccepting (ω^+^) powers and net electrophilicity (Δω^±^), were calculated to evaluate the reactivity of the molecules [40,41].

### 3.5. Ring Current Densities Calculations

The current densities were computed using the GIMIC program [42], which employs the gauge, including the atomic orbital (GIAO) method [43] at the PBE0-D3/Def2-TZVP level. These calculations consider an external magnetic field perpendicularly directed to the molecular plane. In our analysis, diatropic (aromatic) and paratropic (antiaromatic) ring currents circulate clockwise and counterclockwise, respectively. To visualize the currents, we used Paraview 5.10.0 software [44,45]. The ring current strength (RCS) was obtained after considering different rectangular integration planes and placed perpendicular to the ring plane, intersecting the selected bonds, and with a starting point in the center of the rings of interest. For integration, GIMIC uses the two-dimensional Gauss–Lobatto algorithm [18]. Positive or negative RCS values characterize diatropic (aromatic) or patratropic (antiaromatic) net ring currents. In addition, values close to zero suggest a non-aromatic character [21].

### 3.6. Non-Covalent Interactions

The non-covalent interaction index (NCI) was calculated for each representative conformation (200 ns simulation) obtained from the cluster analysis. Non-covalent interactions such as: hydrogen bonds, steric repulsion and van der Waals interactions, were identified and mapped using the promolecular densities (ρ^pro^), calculated as the sum of all atomic contributions. The NCI is based on the electron density (ρ), its derivatives and the reduced density gradients (s). The reduced density gradient is given by:(1)s=123π213 ∇ρρ43

These interactions are local and manifest in real space as reduced gradient isosurfaces with low densities that are interpreted and colored according to the corresponding values of the sign (λ2)ρ. The surfaces are colored on a blue–green–red scale according to the strength and type of interaction. Blue indicates strong attractive interactions; green indicates weak van der Waals interactions, and red indicates strong unbound superposition. All calculations were performed with NCIPlot software [46].

## 4. Conclusions

The binding of nitro compounds to XenB requires deactivating groups (NO_2_ or NH_3_) at the ortho or para position; however, this binding is not sufficient for the enzyme to process the substrate. As observed in molecular dynamics simulations, the Tyr65 residue may be critical for ligand binding to the XenB substrate binding site for all ligands studied and the Tyr335 residue for DN4 and TNB. Electrostatic potentials show that compounds that bind XenB have a large positively charged region surrounding the aromatic ring and including the N atoms of the substituents. The aromatic nitrocompounds TNT (14.7 ev.) and TNB (15.6 ev.) against which XenB shows the highest relative activity show the highest net electrophilicity. All compounds tested and processed by XenB show similar overall hardness (between 2.2 and 2.8 ev.), which does not correlate with the different activity of XenB by these compounds. The more aromatic compounds, with ring current densities on the order of 12.0 nAT^−1^, such as 2AD and NBZ, show high stability, which is consistent with the inactivity of XenB by these compounds. From these results, it is possible to explain the activity of XenB by the tested compounds, which could be used to estimate its activity by other nitro compounds in future studies.

## Figures and Tables

**Figure 1 ijms-24-00400-f001:**
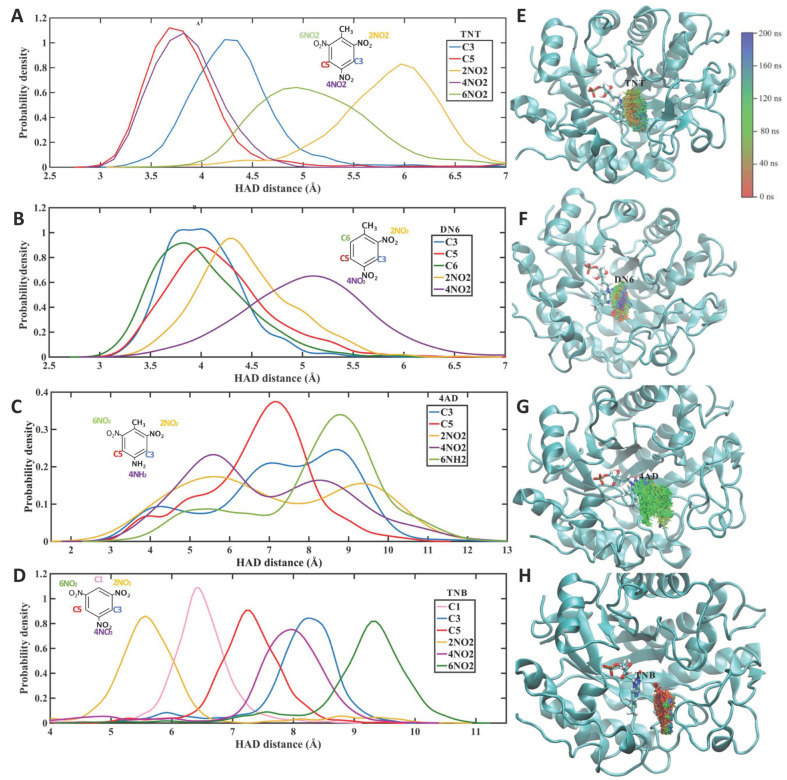
Probability of the HAD distance and position of XenB and the ligands TNT (**A**,**E**), DN6 (**B**,**F**), 4AD (**C**,**G**), and TNB (**D**,**H**). The HAD distance was measured from nitrogen 5 of the FMN to the nitrogen of a nitro group of the ligand (yellow, violet, or green line) or to carbon 3 (blue line), carbon 5 (red line) and carbon 6 of the aromatic ring of the substrate (green line). The nitro groups were named according to the carbon to which they are attached: 2NO_2_ is the nitro group attached to carbon 2 (yellow line), 4NO_2_ is the nitro group attached to carbon 4 (purple line), and 6NO_2_ is the nitro group attached to carbon 6 (green line). NH_2_ groups are named similarly, according to the carbon number of the aromatic ring to which they are attached. 4NH_2_ is the amino group attached to carbon 4 (violet line, panel C). The structures to the right of the plots show the position of the ligand TNT (**E**), DN6 (F), 4AD (**G**) and TNB (**H**) every 1 ns of the 200 ns molecular dynamics simulation. At the top right is the color scheme with the substrate labeled at each simulation time (red for the initial time and blue for the 200 ns simulation).

**Figure 2 ijms-24-00400-f002:**
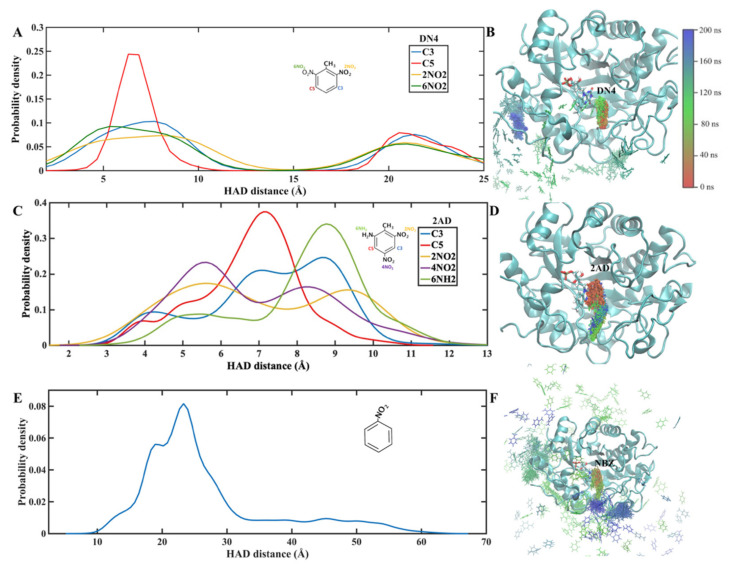
Probability of HAD distance and the positions of ligands that are not processed by XenB. The HAD distance measured from nitrogen 5 of the FMN cofactor to each reducible group are indicated by different colors (**A**,**C**,**E**). The distance to the aromatic ring carbons is observed with a blue line for carbon 3 (C3) and a red line for carbon 5 (C5) (**A**,**C**). The HAD distance to nitro groups attached to carbon 2 (2NO_2_) with a yellow line, for nitro groups attached to carbon 4 (4NO_2_) a purple line, and for nitro groups attached to carbon 6 (6NO_2_) a green line (**A**,**C**). In the case of 2AD (**C**), the green line indicates the HAD distance to the amino group at position 6 (6NH2). For NBZ only the HAD distance to the single nitro group (blue line) is shown. The structures on the right of the graphs show the position of the ligand DN4 (**B**), 2AD (**D**) and NBZ (**F**) every 1 ns of the 200 ns molecular dynamics simulation. At the top right is the color scheme with which the substrate was labeled at each simulation time (red for the initial time and blue for 200 ns of simulation).

**Figure 3 ijms-24-00400-f003:**
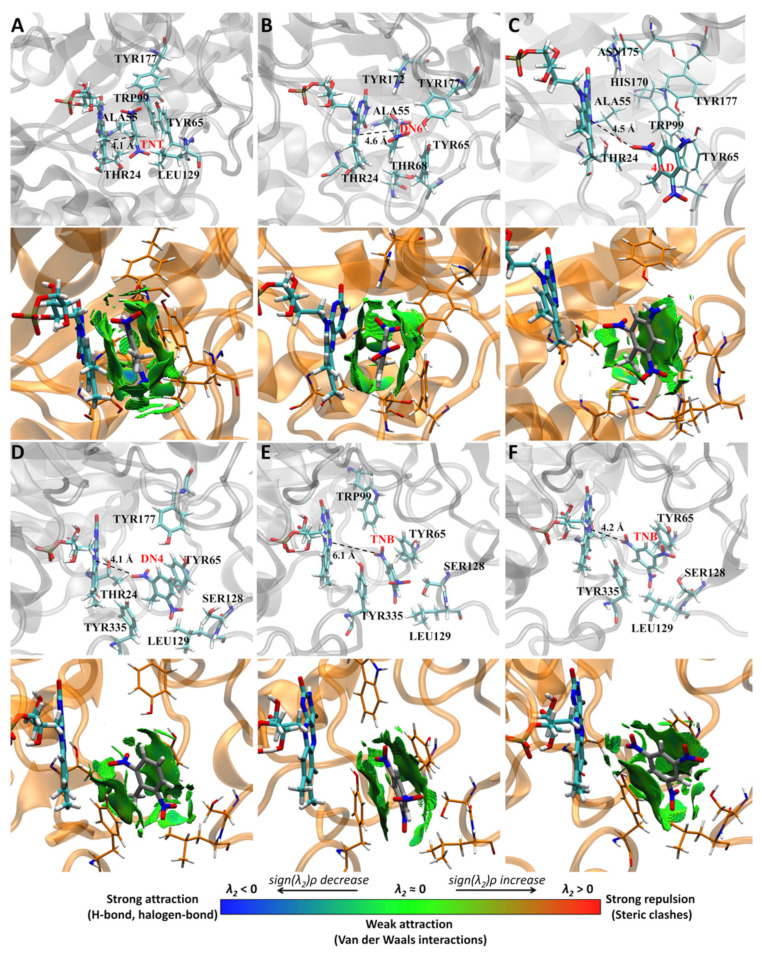
Clustering analysis of molecular dynamics simulation. Application density-based spatial clustering (DBSCAN) algorithm was used to perform cluster analysis of 200 ns molecular dynamics simulations of XenB complexes with TNT (**A**), DN6 (**B**), 4AD (**C**), DN4 (**D**) and TNB (**E**,**F**). Figures (**E**) and (**F**) correspond to two clusters obtained for the XenB/TNB simulation analysis. For each structure, XenB residues within 3 Å of each ligand were highlighted.

**Figure 4 ijms-24-00400-f004:**
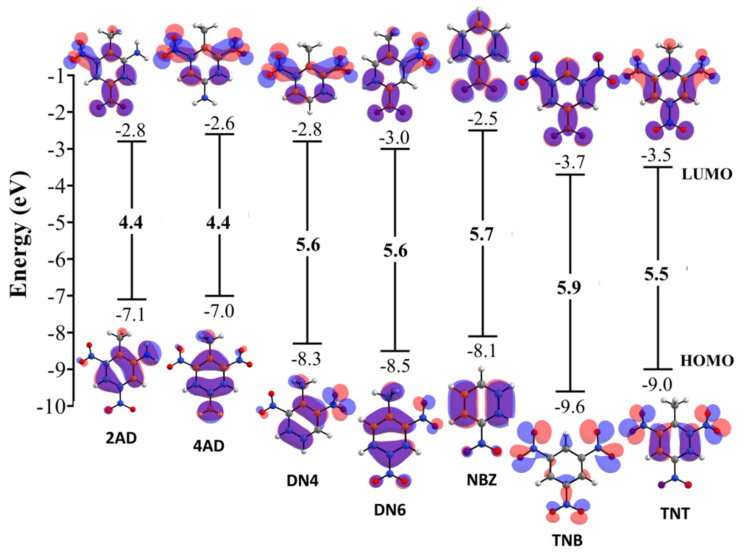
Schematic representation of the HOMO–LUMO energy gaps for respective nitroaromatic compounds and interfacial plot of the orbitals. Blue and red parts of the interfacial plot refer to the different phases of the molecular wave functions, for which the isovalue is 0.03 au.

**Figure 5 ijms-24-00400-f005:**
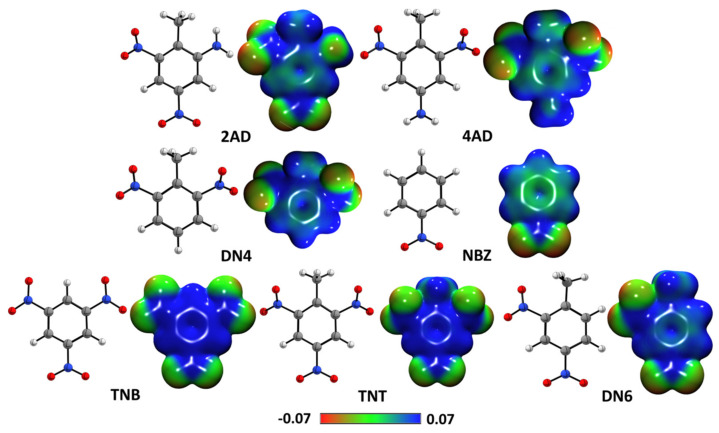
Molecular electrostatic potential maps (in a.u.) of nitroaromatic compounds.

**Figure 6 ijms-24-00400-f006:**
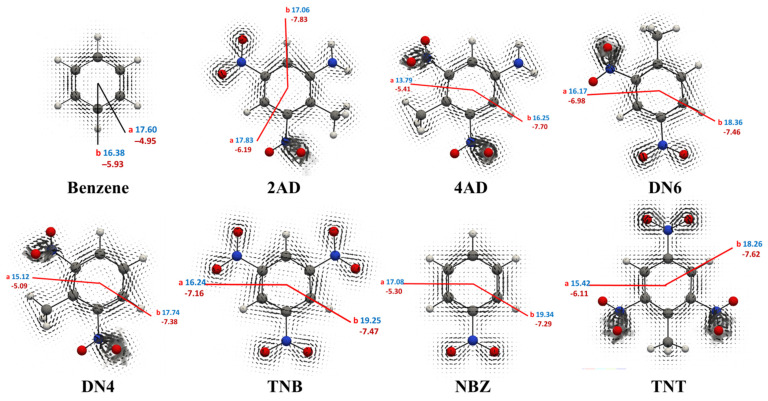
Schematic representation of the ring currents of benzene and seven nitroaromatic compounds. The vector plots are visualized in a plane placed 0.5 Å above the ring plane. The numbers present the RCS values of each ring’s current path in nAT^−1^. Hydrogen (white), carbon (gray), oxygen (red) and nitrogen (blue) atoms are seen as spheres.

**Table 1 ijms-24-00400-t001:** Relative activity of XenB in the presence of different substrates (six nitroaromatic compounds and TNT). The relative activity was calculated in relation to the specific activities (IU/mg protein) of each compound relative to the specific activity of XenB per TNT (100% equals 1.55 U/mg protein min, light blue shading) [13]. The chemical structure of each compound is shown in the second column.

	Estructure	% Activity
2,4,6-Trinitrotoluene (TNT)	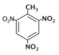	100
2,4-Dinitrotoluene (DN6)	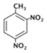	3.7
2,6-Dinitrotoluene (DN4)	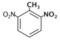	0.2
4-Amino-2,6-dinitrotoluene (4AD)	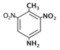	24
2-Amino-4,6-dinitrotoluene (2AD)	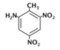	0
1,3,5-Trinitrobenzene (TNB)	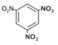	426
Nitrobenzene (NBZ)	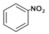	0
		[13]

**Table 2 ijms-24-00400-t002:** Global reactivity descriptors for the nitroaromatic compounds by xenobiotic reductase XenB, calculated with the PBE0-D3 density functional. All units measured in eV.

Compound	$\epsilon_{HOMO}$	$\epsilon_{LUMO}$	$\Delta \epsilon_{HOMO-LUMO}$	*χ*	*η*	*ω*	*ω* ^−^	*ω* ^+^	Δ*ω*;^±^
2,4,6 Trinitrotolueno (TNT)	−9.0	−3.5	5.5	6.2	2.7	7.0	10.5	4.3	14.7
2,4 Dinitrotolueno (DN6)	−8.5	−3.0	5.6	5.8	2.8	5.9	9.1	3.4	12.5
2,6 Dinitrotolueno (DN4)	−8.3	−2.8	5.6	5.6	2.8	5.6	8.7	3.1	11.8
4-amino-2,6-Dinitrotolueno (4AD)	−7.0	−2.6	4.4	4.8	2.2	5.3	7.9	3.1	11.1
2-amino-4,6-Dinitrotolueno (2AD)	−7.1	−2.8	4.4	4.9	2.2	5.6	8.3	3.4	11.7
1,3,5-trinitrobenzeno (TNB)	−9.6	−3.7	5.9	6.6	3.0	7.4	11.1	4.5	15.6
Nitrobenzeno (NBZ)	−8.1	−2.5	5.7	5.3	2.8	5.0	8.0	2.7	10.6

**Table 3 ijms-24-00400-t003:** Equations for global reactivity indices calculated in the TAFF^6^ pipeline.

	Koopmans’ Theorem	Reference
Global Hardness (*η*)	$\eta =\frac{1}{2}\left(\epsilon_{L}-\epsilon_{H}\right)$	[7,8,9,10,11,12]
Electronegativity (*χ*)	$\chi =-\frac{1}{2}\left(\epsilon_{L}+\epsilon_{H}\right)$	[8,13,14]
Electrophilicity (*ω*)	$\omega =\frac{\mu^{2}}{2\eta }=\frac{{\left(\epsilon_{L}+\epsilon_{H}\right)}^{2}}{2\left(\epsilon_{L}-\epsilon_{H}\right)}$	[15]
Electron Acceptor (*ω*^+^)	$\omega^{+}=\frac{{\left(\epsilon_{L}+3\epsilon_{H}\right)}^{2}}{16\left(\epsilon_{L}-\epsilon_{H}\right)}$	[15]
Electron Donator (*ω*^−^)	$\omega^{-}=\frac{{\left(3\epsilon_{L}+\epsilon_{H}\right)}^{2}}{16\left(\epsilon_{L}-\epsilon_{H}\right)}$	[15]
Net Electrophilicity (Δ*ω*^±^)	$\Delta \omega^{\pm }=\omega^{+}\;+\;\omega^{-}$	[16]

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
