# Peer review of "Structural Factors That Determine the Activity of the Xenobiotic Reductase B Enzyme from Pseudomonas putida on Nitroaromatic Compounds"

_ijms, 2022, doi:10.3390/ijms24010400_

Round 1

Reviewer 1 Report

This is an interesting piece of research where molecular dynamics and quantum chemistry is used to explain enzyme activity towards the aromatics rings with nitro functional groups. The methodologies used here are sound and adequate for this study. The relation between the stability of aromaticity and enzyme activity will be very compelling for the readers.

Please consider minor corrections of the entire manuscript, such as-

1. Correction of the subscripts for chemical symbols throughout the manuscript. Example, caption of figure 1  

2. Caption of figure 2 should be in English

Author Response

Dear reviewer, thank you for your comments and corrections. In the manuscript file I highlight the modifications in yellow and in the resp.docx file the responses in red.

This is an interesting piece of research where molecular dynamics and quantum chemistry is used to explain enzyme activity towards the aromatics rings with nitro functional groups. The methodologies used here are sound and adequate for this study. The relation between the stability of aromaticity and enzyme activity will be very compelling for the readers.

Thank you for your review and comments, we believe that the corrections will allow a considerable improvement of our work.

Please consider minor corrections of the entire manuscript, such as

  1. Correction of the subscripts for chemical symbols throughout the manuscript. Example, caption of figure 1

R: Thank you for your indication, according to this, the subscripts of all chemical compounds in the text have been corrected.

  1. Caption of figure 2 should be in English

R: Please accept apologies for this error, I have corrected the language in the caption of Figure 2.

Reviewer 2 Report

See the attachment

Author Response

Dear reviewer, thank you for your comments and corrections. In the manuscript file I highlight the modifications in yellow and in the resp6dic.docx file the responses in red.

Major concerns

  1. The authors stated that “If one considers the reaction catalyzed by ..., the promiscuity of substrates may be consistent with conformational flexibility of the catalytic site ...”. In my opinion, the catalytic site is composed of catalytically critical residues, whose flexibility/mobility unlikely influences the substrate specificity/promiscuity because of the very similar mobility of the catalytic sites among the individual members within an enzyme family.

R: I thank you for the comment and the possibility to discuss these topics on structural dynamics and catalytic activity. I agree with your first statement about catalytic residues; however, I think it is likely that there is an interaction between flexibility/mobility and substrate binding, and therefore that this affects the specificity/promiscuity of XenB. My assertion is supported by work on the flexibility of homologous proteins belonging to the same family. The substrate binding region in the CYP2C9 enzyme is more flexible than in CYP101, which reduces the size of the binding pocket and the ligand entry channel to the catalytic site (Figure 3)[1]. In this work they do not test other ligands, however, a stiffer binding site, smaller volume and narrower entry channel could make the selection more stringent. Although for proteins of the same family, the main regions with different flexibility are not in the catalytic site[2], it is possible to identify paralogous enzymes with catalytic sites with different flexibility and affinity for the substrate. An example of this is the temperature adaptation observed in the enzyme lactate dehydrogenase from marine mollusks and in endonuclease I, which shows that this is a probable phenomenon in several families of enzymes[3].

According to the discussion developed, the references cited in the answer at the end of this paragraph will be added "If one considers the reaction catalyzed by XenB and using a single cofactor (FMN), the promiscuity of substrates may be consistent with conformational flexibility of the catalytic site, substrate ambiguity, different protonation states or different subsites within the active site".

  1. The phrases like “specificity/promiscuity of this enzyme”, “selectivity or promiscuity of XenB”, “enzyme promiscuity”, etc. is misleading. The authors need to distinguish between “enzyme’s substrate specificity/promiscuity” and “enzyme’s specificity/promiscuity”.

R: Thanks for the suggestion. We will clarify that idea by focusing on substrate promiscuity, understanding it as a residual catalytic capacity of the enzyme for other substrates. According to this approach we will modify the sentences including "specificity/promiscuity of this enzyme", "selectivity or promiscuity of XenB", "promiscuity of the enzyme" by substrate promiscuity as appropriate. The following lines were modified:

Line 23: "specificity/promiscuity of this enzyme" for " the substrate promiscuity of this enzyme ".

Line 32 keyword was modified "specificity/promiscuity" for "substrate promiscuity of enzymes"

Line 41 was added substrate

Line 43 was replaced "enzyme promiscuity" for " substrate promiscuity of enzymes"

Line 45 was replaced "enzyme promiscuity" for " this enzymatic capacity "

Line 56 was modified text for "substrate promiscuity of enzymes"

Line 69 was added "of XenB"

Line 76 was modified to "substrate promiscuity of XenB"

  1. The described panel labels in the title of Figure 1 are wrong. For example, “ligands TNT (A and E)” should be “ligands TNT (A and B)”, “DN6 (B and F)” should be “DN6 (C and D)”, etc. There are also label errors in legend of Figure 1. For example, “4NH2 is the amino group attached to carbon 4 (green line, panel C). The structures to the right of the plots show the position of the ligand TNT (E), DN6 (F), 4AD (G) and TNB (H)”. In Figures 1E, some of the labels for groups are wrong. For example, “4NO2” should be “4NH2”, “6NH2” should be “6NO2”.

R: the errors in the legend of figure 1 were corrected.

  1. The legend of Figure 2 was not written in English.

R: added the following legend in English.

  1. The authors may explain why “the negative regions (green/red) indicate the regions that will perform electrophilic attacks”. In my opinion, the negative regions would suffer an electrophilic attack or perform a nucleophilic attack.

R: Dear reviewer, the error in that sentence has been amended, as it was a typing error. The final sentence is now in the manuscript.

” The positive regions (blue/green) indicate the portion of the molecule with the higher probability to suffer a nucleophilic attack, while the negative regions (green/red) indicate the regions that will perform nucleophilic attacks [22], that is, the chemical reactivity of molecules since regions of negative potential are expected to be sites of protonation and nucleophilic attack while regions of positive potential may indicate electrophilic sites.”

  1. Why did the authors state that “... in the case of TNB and TNT, the nitrogen atom of the nitro group attracts electrons, ...”? According to “Figure 2” (which should be Figure 5), it is can be clearly observed that it is the oxygen atom rather than the nitrogen atom of the nitro group that attracts electrons. In addition, the expression “... generates electronic deficiency in the aromatic ring, favoring nucleophilic attack” is misleading because the aromatic ring with the strong positive electrostatic potential favor the reaction of suffering a nucleophilic attack, instead of performing a nucleophilic attack. Another question is why the authors said that the cation/pi interaction could further stabilize the enzyme/substrate complex. Could the author provide a figure showing the so-called cation/pi interactions?

This error in the first writing has been corrected in the new text. The phrase "nitrogen atom" has been changed to "oxygen atom" to be consistent with what is observed in Figure 5 (which has also been corrected). In addition, we assume an error in the wording when translating the writing. The paragraph in question has been modified in lines 322-325 by the following:

“It is worth mentioning that in the case of TNB and TNT, the oxygen atom of the nitro group attracts the electron density of the ring, generating electronic deficiency, which favors the ring to suffer a nucleophilic attack”

Another question is why the authors said that the cation/pi interaction could further stabilize the enzyme/substrate complex. Could the author provide a figure showing the so-called cation/pi interactions?

R: We have not performed calculations to test this hypothesis, so this statement is speculative. Therefore, we will add this sentence in line 317 to 321 ". However, to evaluate this possibility, quantum calculations describing the interaction of Tyr54 and each substrate at a higher level of theory are required, which we may address in future work."

  1. The methods used were not well described. The authors should describe which software was used to modify the function groups of TNT in 4AEO. Is the “downward minimization” the “steepest descent minimization”? “Langev dynamics” should be “Langevin thermostat”. How much is the relaxation time for the Berendsen barostat? The reference 17 is an inappropriate citation for Langevin thermostat and Berendsen barostat. The original works of the two algorithms (Langevin thermostat and Berendsen barostat) need to be cited. In addition, the reference for the PME method should also be cited. The “the density-based spatial method of applications with noise (DBSCAN)” should be “density-based spatial clustering of applications with noise (DBSCAN)”. Reference 30 is only for CPPTRAJ. The work of DBSCAN needs also to be cited. The description of DBSCAN in the Method section differs from that in the Result section.

R: All the modifications indicated in detail were carried out as follows

  1. a) The authors should describe which software was used to modify the function groups of TNT in 4AEO

The following sentence was added on line 357 “the Avogadro program was used to modify the coordinates by removing or adding functional groups”.

  1. b) Is the “downward minimization” the “steepest descent minimization”?

The term was corrected, and the structural minimization protocol was detailed. After the modifications, the sentence of line 374 remained as follows “minimization and structural relaxation of water molecules with 1000 of steps steepest descent minimization, followed by 1000 of steps conjugate gradient minimization”. Similarly, the polypeptide chain minimization steps were specified on line 378.

  1. c) “Langev dynamics” should be “Langevin thermostat”

R: were corrected

  1. d) How much is the relaxation time for the Berendsen barostat?

R: Added Berensen's Borostat relaxation time in line 386 by including the following phrase “with 2 ps of relaxion time”

  1. e) The original works of the two algorithms (Langevin thermostat and Berendsen barostat) need to be cited. In addition, the reference for the PME method should also be cited.

The following references were added 29, 30 y 31 to cite original works.

  1. f) The “the density-based spatial method of applications with noise (DBSCAN)” should be “density-based spatial clustering of applications with noise (DBSCAN)”. Reference 30 is only for CPPTRAJ. The work of DBSCAN needs also to be cited. The description of DBSCAN in the Method section differs from that in the Result section

Line 393 has been corrected by adding “density-based spatial method of applications with noise”, reference 34 was added and the sentence on line 178 was modified to correct the error in the results of the description of this methodology. Line 181 was “without considering the ligand hydrogen atoms and at least 5 points for each cluster.”

Minor concerns

  1. The sentence “However, structural factors to explain the binding of different substrates to XenB” is incomplete.

R: was reorganized and completed the sentence on line 20 “the structural factors that explain the binding of XenB to different substrates are unknown”

  1. The key words “Selectivity/promiscuity enzyme activity” and “Interaction ligand enzyme prediction” are inappropriate.

R: were corrected

  1. The references for “A theoretical study on the reduction of TNT and the Meisenheimer complex ...”, “XenB shows activity on the benzene compounds TNT, DN6, 4AD and TNB, but not for DN4, 2AD or NBZ” need to be cited.

R: were cited with 18 and 13 reference

  1. “As shown in Figure 2, the functional groups closest to ...” should be “As shown in Figure 1, the functional groups closest to ...”.

R: were corrected

  1. Where can I find Figure 1H?

R: is in the lower left corner of Figure 1. It corresponds to the structure of XenB in complex with TNB.

  1. The authors stated that “... residues Tyr 65 and Leu129 remain within 3 Å of substrates TNT (A), DN6 (B), 2AD (C), DN4 (D) and TNB (E and F)”. However, I cannot find Leu129 in Figures 3B and C, and “2AD” should be “4AD”.

R: The error was corrected by arranging the sentence as follows (lines 200-203) “As observed in this analysis, the Tyr 65 residue remains within 3 Å of the substrates TNT (A), DN6 (B), 2AD (C), DN4 (D) and TNB (E and F) and Leu129 remains a short distance (less than 3 Å) from TNT, DN4 and TNB, establishing weak interactions (green marking in the lower panels) with these substrates”

  1. What do ICNs mean in “the nature of the ICNs”?

R: The error was corrected, it should read NCIs

  1. “Trip99” should be “Trp99”

R: were corrected in 217 line

  1. What are the differences between nitroaromatic compounds and nitramines?

R: The error was corrected by eliminating nitroamines and specifying the compounds analyzed by adding "(TNT, DN6, DN4, 4AD, 2AD, TNB and NBZ" in line 253.

  1. Please consider to rephrase the following sentence to convey clearly what you want to say. “The boundary orbitals p information about the energies of the Highest Occupied Molecular Orbital (HOMO) and the Lowest Unoccupied Molecular Orbital (LUMO), as well as acting as electron donors and acceptors, respectively[21].”

R: was rephrase to “The p-boundary orbitals provide information about the energies of the Highest Occupied Molecular Orbital (HOMO) and the Lowest Unoccupied Molecular Orbital (LUMO), that can acting as electron donors and acceptors, respectively.”

  1. “Table 2 and Figure 1 show the values of HOMO, ...”, in which Figure 1 should be Figure 4.

R: were corrected

  1. Please rephrase the conclusive sentence “These potential and charge differences around the compounds could be the main responsible for the stabilization between the XenB complexes and for part of their reactivity” to accurately convey what you want to say.

R: was rephrase to “The positive charge distribution around the aromatic ring could favor the binding between the studied complexes and the catalytic site of XenB, stabilizing the enzyme/substrate complex.”

  1. “... are useful in measure the ...” should be “... are useful in measuring the ...”

R: was rephrase and corrected sentence.

  1. “despite of their values are shorter than benzene” is a grammatically incorrect statement.

R: was rephrase the sentence to “The RCS value obtained with the same level of theory shows the aromatic character of nitrobenzene, although weaker than for benzene, which is deduced from its more positive paratropic ring currents”

  1. Change “The initial coordinates of the P. putida XenB enzyme in complex with TNT and the cofactor FMN were obtained from the published crystal structure in the Protein Data Bank 4AEO database at a resolution of 1.8 Å” to “The crystal structure of the P. putida XenB enzyme in complex with TNT and the cofactor FMN was obtained from the Protein Data Bank (PDB) with PDB ID: 4AEO at a resolution of 1.8 Å”. In addition, the paper of this structure should be cited.

R: was corrected, but I cannot add the quote as the structure has not yet been published.

  1. The unit for the harmonic constraints is wrong, which is kcal/mol/Å2, instead of kJ/mol Å2.

R: The AMBER program works with Kcal units for energy, as you can check in the AMBER program manual22 (page 385) with the command "restraint_wt".

  1. Please rephrase the sentence “... isothermalisobaric assembly (NPT) without constraints for 200 ns at 300 Ky 1 atm ...”

R: was rephrase to “For production, molecular dynamics simulations were run for 200 ns with a 2 fs time step, unconstrained, in an isothermal-isobaric (NPT) assembly at 300 K”

Reference

  1. Huang, J.; Xu, Q.; Liu, Z.; Jain, N.; Tyagi, M.; Wei, D.Q.; Hong, L. Controlling the Substrate Specificity of an Enzyme through Structural Flexibility by Varying the Salt-Bridge Density. Molecules 2021, 26, doi:10.3390/molecules26185693.
  2. Richard, J.P. Protein Flexibility and Stiffness Enable Efficient Enzymatic Catalysis. J Am Chem Soc 2019, 141, 3320-3331, doi:10.1021/jacs.8b10836.
  3. Dong, Y.W.; Liao, M.L.; Meng, X.L.; Somero, G.N. Structural flexibility and protein adaptation to temperature: Molecular dynamics analysis of malate dehydrogenases of marine molluscs. Proc Natl Acad Sci U S A 2018, 115, 1274-1279, doi:10.1073/pnas.1718910115.

Round 2

Reviewer 2 Report

See the attachment

Author Response

Thank you for the opportunity to improve our work and for the dedication shown. Below we respond point by point to your corrections, in addition to a general correction of the English wording.

Although some of my previous concerns were addressed in the revised version, there are still many errors in the current manuscript, and the English writing needs to be overall checked for a substantial improvement. I stick to my previous recommendation that manuscript be rejected with the possibility to resubmit.

  1. Please check the sentence “XenB can reduce the nitro groups or the aromatic ring of 2,4,5‐trinitrotoluene (TNT) or 2,4,5‐trinitrophenol (TNF)”. “2,4,5” may be “2,4,6”

R: the nomenclature error of the compound TNT in lines 25, 67 and 60 has been corrected, the correct name is 2,4,6-trinitrotoluene and 2,4,6-trinitrophenol.

  1. The catalytic site, although possibly overlaps with the substrate‐binding site/region, differs from the substrate‐binding site. The main difference between the substrate‐binding site and catalytic site is that the binding site temporarily binds with the substrate whereas the catalytic site catalyzes the reaction of the substrate. Therefore, the term “catalytic site” in the paper needs to be replaced by “substrate‐binding site” or “binding site”

R: Thank you for your comment and according to what you indicate, lines 62, 65, 91, 115, 209, 462 have been corrected by replacing "catalytic site" by "substrate binding site". This modification makes it possible to refer generally to the region that binds the ligand.

  1. The panel labels (i.e., A, B, C, etc) were changed in Figure 1 of the current version, but the descriptions of these labels are wrong. For example, “TNT (A and B)” should be TNT (A and E).

R: the legend in figure 1 has been corrected

  1. Please rephrase the sentence “In yellow they HAD distance to the nitro group attached to carbon 2 (2NO2), with a purple line carbon 2 (4NO2), with a purple line the HAD distance to the nitro group attached to carbon 4 (4NO2) and a green line the HAD distance to the nitro group on carbon 6 (6NO2)” to convey clearly what you want to say. There are description errors in the sentence “For the case of 4AD (C), the HAD distance to the amino group at position 4 is indicated with the green line (6NH2).” I cannot find out “the position of the ligand TNT (E), DN6 (F), 4AD (G) and TNB (H)” in Figure 2.

  1. a) Please rephrase the sentence “In yellow they HAD distance to the nitro group attached to carbon 2 (2NO2), with a purple line carbon 2 (4NO2), with a purple line the HAD distance to the nitro group attached to carbon 4 (4NO2) and a green line the HAD distance to the nitro group on carbon 6 (6NO2)” to convey clearly what you want to say

R: the paragraph was reworded in lines 159 to 165 to " The HAD distance measured from nitrogen 5 of the FMN cofactor to each reducible group are indicated by different colors. The distance to the aromatic ring carbons is observed with a blue line for carbon 3 (C3) and a red line for carbon 5 (C5). The HAD distance to nitro groups attached to carbon 2 (2NO2) with a yellow line, for nitro groups attached to carbon 4 (4NO2) a purple line and for nitro groups attached to carbon 6 (6NO2) a green line.  ".

  1. b) There are description errors in the sentence “For the case of 4AD (C), the HAD distance to the amino group at position 4 is indicated with the green line (6NH2).”

R: The error on position 4 to position 6 was corrected “For the case of 4AD (C), the green line indicates the HAD distance to the amino group at position 4 (6NH2)”

  1. c) I cannot find out “the position of the ligand TNT (E), DN6 (F), 4AD (G) and TNB (H)” in Figure 2.

R: The error in the legend has been corrected. This referred to the ligands in Figure 1 TNT (E), DN6 (F), 4AD (G) and TNB (H). Therefore, the legend of Figure 2 was modified to read "The structures on the right of the graphs show the position of the ligand DN4 (B), 2AD (D) and NBZ (F)" on lines 165 and 166.

  1. The authors said “However, to evaluate this possibility, quantum calculations describing the interaction of Tyr54 and each substrate at ...”. Does it mean that only Tyr54 can form the potential pi/pi or cation/pi interactions with TNB and TNT? If so, the authors may explain why? Is there any structural evidence supporting this?

  1. a) The authors said “However, to evaluate this possibility, quantum calculations describing the interaction of Tyr54 and each substrate at ...”

R: for the system analyzed only Tyr54 could establish relevant interactions of this type with all ligands during the simulation. This statement does not exclude some short-lived interaction between 4AD and Trip99.

  1. b) Does it mean that only Tyr54 can form the potential pi/pi or cation/pi interactions with TNB and TNT?

R: According to our simulations, only Tyr54 would be at a short enough distance for a pi/pi or cation/pi interaction with TNB and TNT to be possible.

  1. c) If so, the authors may explain why? Is there any structural evidence supporting this?

R: This type of interaction is important for secondary structure stabilization and there is literature supporting the possibility that it stabilizes ligand binding. However, as mentioned in the previous response, this statement is speculative, and more background is needed to support it. Furthermore, since it is not necessary to establish the conclusion and only suggests studies that could give continuity to our work, we eliminate it from the discussion. We deleted the paragraph “This charge distribution could further stabilize the enzyme/substrate complex through pi/pi or cation/pi interactions, although the positive charge is delivered by the dipole of the ring of each substrate. However, to evaluate this possibility, quantum calculations describing the interaction of Tyr54 and each substrate at a higher level of theory are required, which we may address in future work. The positive charge distribution around the aromatic ring could favor the binding between the studied complexes and the substrate‐binding site of XenB, stabilizing the enzyme/substrate complex”.

  1. I cannot read what the author wrote in the line 338.

R: We apologize for that mistake in the previous version. This was corrected in the revised version of the ms.

  1. The reference for the “Avogadro program” needs to be cited.

R: OK, the requested reference was included (ref. 28), line 365.

  1. “1000 of steps ...” should be “1000 steps of ...”; “NPT assembly” should be “NPT ensemble”; “canonical assembly (NVT)” should be “canonical ensemble (NVT)”

R: OK, as requested this was corrected in the new version of the ms. (see new lines 381, 382 and 385).

  1. The unit of harmonic constraints, kcal/molÅ2 is misleading, it should be replaced by kcal/mol/Å2 or kcal/(mol∙Å2).

R: OK. As requested, the units of harmonic constraints were corrected in the new version of the ms (see new line 384).

  1. I cannot read what the author wrote in the line 403.

R: We apologize for this error in the previous version. It has been corrected in the revised version of the ms on lines 3689 to 301 to read " 5) for production, molecular dynamics simulations were run for 200 ns with a 2 fs time step, unconstrained, in an isothermal-isobaric (NPT) ensemble at 300 K. In the MD simulation, temperature was controlled with Langevin thermostat with a collision frequency of 1 ps-1 (NVT)"

  1. I cannot understand why the reference 13 was cited for “in this work we evaluated the structural factors determining the interaction of XenB and the nitrocompounds TNT, DN4, DN6, 4AD, 2AD, TNB, and NBZ[13]”.

R: The reference has been deleted, since it refers to the work carried out in this research.

  1. In the sentence “As seen in Figure 1, .... the 200 ns of simulation (Figure 1B, 1D, 1F and 1H)”, the figure citations are wrong.

R: The sentence "As seen in Figure 1" was corrected and the paragraph on lines 209 and 216 was modified to clarify this point. The final paragraph was left as indicated: “Figure 3 shows the representative conformations from the 200 ns molecular dynamics simulation for the enzyme/ligand complexes that can be processed by XenB: TNT (Figure 3A), DN6 (Figure 3B), 4AD (Figure 3C), DN4 (Figure 3D) and TNB (Figure 3E and 3F). These representative conformations were analyzed by QM calculations to identify the nature of the NCIs formed between the protein and each ligand (bottom panels Figure 3)”

  1. I cannot find 2AD in Figure 3C.

R: OK. The error in the legend was corrected, modifying line 203 to read " molecular dynamics simulations of XenB complexes with TNT (A), DN6 (B), 4AD (C),"

  1. There are still the noun phrase “nitramine explosive compounds” in the text.

R: OK. We deleted " nitramine explosive " in ms.

  1. “that can acting as ...” is grammatically incorrect.

R: was changed to "which can act as” in line 273 to rad “The p-boundary orbitals provide information about the energies of the Highest Occupied Molecular Orbital (HOMO) and the Lowest Unoccupied Molecular Orbital (LUMO), which can act as electron donors and acceptors, respectively”

  1. The authors stated that “... favor the binding between the studied complexes and the catalytic site of XenB ...”. It should be noted that the complex is an entity composed of the protein and ligand or the enzyme and substrate. I cannot understand how the studied complexes bind to the catalytic site of XenB.

R: OK. The sentence was deleted

Finally, the sentence in lines 462 and 463 has been corrected to specify the possible participation of each waste. The modified sentence reads as follows "As observed in molecular dynamics simulations, the Tyr65 residue of XenB may be critical for ligand/enzyme complex formation for all compounds studied, whereas the Tyr335 residue only for DN4 and TNB"

A paragraph explaining the difference between substrate promiscuity and substrate ambiguity has also been added, in the added paragraph is " The ability of an enzyme to process different substrates has been termed promiscuity and is a property identified in several enzymes. It is a relevant ability since it could be associated with the evolution of proteins. For the case of chemically related substrates, such as the case we will address in this article, it has been called substrate ambiguity. Thus, when we speak in a general way about enzymes that can process different substrates, we will say substrate promiscuity and when we refer to the special case we are dealing with in this article we will say substrate ambiguity." on line 43 to 49, in addition to the reference. Errors in the text were corrected and the English writing was generally revised.

To highlight the applications of the work, the last line of the abstract was modified in line 29, 30 “These results are consistent with experimental data and could be used to propose point mutations that allow this enzyme to process new molecules of biotechnological interest”.

Round 3

Reviewer 2 Report

All my previous concerns have been addressed in the present version. I recommend the acceptance of manuscript after minor revisions. 

When the abbreviation “NCI” first appears in the text, its full term should be provided.

"XenB inactive in presence of ..." should be "XenB is inactive in presence of ..."   The references need to be cited for descriptions in lines 77-83.  

Author Response

Thank you very much for your dedication and understanding in reviewing our work. Based on your comments, we have made the following corrections:

When the abbreviation "NCI" appears for the first time in the text, its full term should be indicated.

R: the sentence on line 212 was modified to add the description of the acronym, leaving the sentence as follows: " These representative conformations were analyzed by QM calculations to identify the nature of the non-covalent interaction index (NCI) which allows describing this type of interaction between the protein and each ligand (bottom panels Figure 3)"

"XenB inactive in the presence of ..." should be "XenB is inactive in the presence of ...".   The references to the descriptions on lines 77-83 need to be cited.

R: Corrected on line 79 in accordance with the final ms to read " XenB is inactive in presence of 2,6-dinitrotoluene (DN4) despite its chemical similarity to TNT or 2,4-dinitrotoluene (DN6)".

The document is attached with change control to follow the indicated modifications.